# Characterization and Spike Gene Analysis of a Candidate Attenuated Live Bovine Coronavirus Vaccine

**DOI:** 10.3390/ani14030389

**Published:** 2024-01-25

**Authors:** Gyu-Nam Park, SeEun Choe, Sok Song, Ki-Sun Kim, Jihye Shin, Byung-Hyun An, Soo Hyun Moon, Bang-Hun Hyun, Dong-Jun An

**Affiliations:** 1Virus Disease Division, Animal and Plant Quarantine Agency, Gimcheon 39660, Republic of Korea; changep0418@korea.kr (G.-N.P.); ivvi59@korea.kr (S.C.); ssoboro@korea.kr (S.S.); kisunkim@korea.kr (K.-S.K.); shinji227@korea.kr (J.S.); msh103101@korea.kr (S.H.M.); hyunbh@korea.kr (B.-H.H.); 2Department of Virology, College of Veterinary Medicine and Research Institute for Veterinary Science, Seoul National University, Seoul 08826, Republic of Korea; anbh5043@snu.ac.kr

**Keywords:** BCoV, vaccine, mutation, CPE, calf

## Abstract

**Simple Summary:**

Diarrheal diseases in calves cause economic losses in countries worldwide. In particular, bovine coronavirus, the main cause of diarrhea in calves, results in high economic losses for cow farmers. In South Korea, calf diarrhea, which affects newborn calves, and winter dysentery, which occurs in adult cows during the winter, are detected continuously. The vaccine strain (BC94) used in South Korea belongs to the GI type; however, a phylogenetic analysis revealed that all of the prevalent circulating strains belong to the GIIa type. Therefore, we attempted to develop a live attenuated BCoV vaccine candidate that targets recent prevalent strains.

**Abstract:**

The bovine coronavirus (BCoV) KBR-1 strain, obtained from calf diarrhea samples collected in 2017, belongs to group GIIa. To attenuate this strain, it was subcultured continuously (up to 79 times) in HRT-18 cells, followed by 80–120 passages in MDBK cells. The KBR-1-p120 strain harvested from MDBK cells at passage 120 harbored 13 amino acid mutations in the spike gene. Additionally, the KBR-1-p120 strain showed a high viral titer and cytopathogenic effects in MDBK cells. Seven-day-old calves (negative for BCoV antigen and antibodies) that did not consume colostrum were orally inoculated with the attenuated candidate strain (KBR-1-p120), or with KBR-1 passaged 10 times (KBR-1-p10) in HRT-18 cells. Calves inoculated with KBR-1-p10 had a low diarrhea score, and BCoV RNA was detected at 3–7 days post-inoculation (DPI). The virus was also present in the duodenum, jejunum, and ileum at autopsy; however, calves inoculated with KBR-1-p120 had low levels of BCoV RNA in feces at 4–6 DPI, and no diarrhea. In addition, an extremely small amount of BCoV RNA was present in the jejunum and ileum at autopsy. The small intestines of calves inoculated with KBR-1-p120 were emulsified and used to infect calves two more times, but pathogenicity was not recovered. Therefore, the KBR-1-p120 strain has potential as a live vaccine candidate.

## 1. Introduction

Bovine coronavirus (BCoV), which possesses a single-stranded, positive-sense, non-segmented RNA genome, is a member of the order *Nidovirales*, family *Coronaviridae* [1]. BCoV is an important pathogen that causes not only neonatal calf diarrhea (NCD) [2], but also winter dysentery (WD) in adult cattle [3] and respiratory disease in cattle of all ages [4,5]. BCoV particles comprise four major structural proteins: the membrane (M) glycoprotein, an envelope (E) protein, a spike (S) glycoprotein, and the hemagglutinin-esterase (HE) glycoprotein [6]. In particular, the S glycoprotein possesses domains responsible for receptor binding and the induction of neutralizing antibodies, which have been exploited for the molecular characterization of isolates [7]. Additionally, the S glycoprotein comprises two subunits: S1 (N-terminal half) and S2 (C-terminal half). Of these, S1 is a globular subunit responsible for virus binding to host cell receptors [8], the induction of neutralizing antibodies [9], and hemagglutinin activity [10]. The S1 hypervariable region is also useful for studying the variability and evolution of BCoV [11,12]. The S1 sequence is diverse, and mutations in this region are associated with changes in antigenicity and viral pathogenicity [13].

WD and NCD have occurred in Korea since the early 2000s [14,15]. Additionally, animal experiments confirmed that BCV, the cause of WD, can induce pathogenic effects in both the digestive and respiratory systems of calves [15]. Recent studies show that BCoVs detected in the diarrheal feces of Korean calves belong to the GIIa genotype, suggesting that there is an antigenic difference from the currently used BC94 (G1 genotype) vaccine strain [16,17]. Thus, it unclear whether the classical BCoV vaccine protects against the recently prevalent GIIa genotype; this is particularly important because as BCoV continues to evolve, new genetic types are being formed and the genes are diversifying. Therefore, it is necessary to develop vaccines that are specific for various BCoV genotypes.

The purpose of this study was to develop a novel live attenuated vaccine candidate through continuous passage of the BCoV isolate prevalent in South Korea, and to conduct a comparative analysis of the S gene. Additionally, a BCoV isolate (KBR-1-p10) and the attenuated candidate strain (KBR-1-p120) were administered orally to calves, after which diarrhea scores, virus shedding, and the presence of viruses in organs were evaluated.

## 2. Materials and Methods

### 2.1. Cell Passage and Indirect Immunofluorescence Assay

This study used a KBR-1 strain isolated from a calf with diarrhea in the Boryeong region (located approximately 150 km south of Seoul) of South Korea [17]. The KBR-1 strain was propagated via inoculation at a multiplicity of infection (MOI) of 0.1 onto monolayers of human rectal tumor (HRT-18) cells cultured in RPMI 1640 (Gibco, Grand Island, NY, USA) containing 5 μg/mL of trypsin (Gibco, Grand Island, NY, USA), followed by incubation at 37 °C/5% CO_2_ for 4 days. Additionally, to increase the titer of the KBR-1 strain, it was inoculated (MOI = 0.1) onto Madin–Darby bovine kidney (MDBK) cells cultured in α-MEM (Gibco, Grand Island, NY, USA) containing 5 μg/mL of trypsin (Gibco, Grand Island, NY, USA), followed by incubation at 37 °C/5% CO_2_ for 4 days to observe cytopathic effects (CPEs). At 4 days post-inoculation of the KBR-1 strain onto HRT-18 or MDBK cells, virus proliferation was confirmed in an indirect immunofluorescence assay (IFA). Briefly, at 4 days post-KBR-1 inoculation, cells were fixed in 80% cold acetone and then reacted for 1 h with a monoclonal antibody specific for BCoV (Absolute Antibody Ltd, Cat no. EGO012, Wilton Centre Redcar, UK). Additionally, after reacting cells for 1 h with FITC-labeled anti-mouse IgG (H + L) adsorbed on human serum (SeraCare Life Sciences, Inc., Cat no. 5230-0307, Milford, MA, USA), fluorescence was observed under a fluorescence microscope.

### 2.2. Phylogenetic Analysis of BCoV

The complete S gene sequences of 12 KBR-1 strains of BCoV, obtained after inoculation onto HRT-18 cells and MDBK cells, were used for phylogenetic analysis. Additionally, the complete S gene sequences of 94 reference BCoV strains were obtained from the National Center for Biotechnology Information (NCBI) GenBank database. Multiple-nucleotide sequence alignments of all 106 BCoV strains were carried out in the CLUSTAL X alignment program (ver. 2.1.) [18], and were analyzed phylogenetically using the MEGA X software [19] using the maximum-likelihood (ML) method, the Tamura–Nei model, and bootstrap analysis (*n* = 1000). The ML tree was constructed using rates among sites (gamma distributed with invariant sites (G + I)) and the ML heuristic method (nearest-neighbor interchange (NNI)).

### 2.3. Comparison of the S Genes of Passaged BCoV Strains

After the inoculation of the BCoV KBR-1 strain into HRT-18 and MDBK cells, total RNA was extracted using the RNeasy mini kit (Qiagen, Cat. No. 74104, Maryland, MD, USA), and cDNA was prepared using the Helix Cript™ Easy cDNA Synthesis Kit (NanoHelix, Daejeon, Republic of Korea). We conducted a PCR of the complete S gene (4092 bp), using six previously published oligo primer sets [20], with amplified fragments of 920 bp, 769 bp, 828 bp, 872 bp, 916 bp, and 653 bp. The six PCR products were sequenced at the Cosmo Genetech Institute (Daejeon, Republic of Korea) using an ABI Prism 3730xl DNA sequencer to generate the complete S gene sequence. Comparative analyses of the complete BCoV S genes were carried out via multiple sequence alignments using Bio Edit Sequence Alignment Editor (version 7.2).

### 2.4. Calf Experiments

To investigate the pathogenicity of the KBR-1-p120 strain (passaged in MDBK cells) and the KBR-1-p10 strain (passaged in HRT-18 cells), 7-day-old calves (negative for BCoV antigens in real-time PCR [21] and negative for antibodies in a serum neutralization antibody test [17]) that did not consume colostrum were assigned to two groups (G1 and G4). Two calves in the G1 group received an oral inoculation of KBR-1-p120 at 10^7.0^ TCID_50_/mL per dose. The G4 group received KBR-1-p10 (concentrated through density gradient ultracentrifugation due to a low titer (10^3.5−4.0^ TCID_50_/mL)) via an oral inoculation of 10^5.0^ TCID_50_/mL per dose. Clinical symptoms (appetite, activity, diarrhea, and nasal) were scored for 7 days after oral inoculation of the KBR-1-p10 and KBR-1-p120 strains. Clinical symptoms were scored as follows: appetite (normal appetite: 0; abnormal appetite: 1); activity (normal activity: 0; abnormal activity: 1); nasal symptoms (normal nasal: 0; rhinorrhea: 1); and diarrhea (no diarrhea: 0; mild diarrhea: 1; moderate diarrhea: 2; and severe diarrhea: 3). Additionally, fecal and nasal samples were collected daily for 7 days post-inoculation (DPI). These samples were collected using sterilized cotton swabs, placed in a liquid transport medium, vortexed, and stored in a deep freezer at −70 °C until real-time PCR (RT-PCR). BCoV RNA was detected via RT-PCR to obtain cycle threshold (ct) values [21]. To examine the pathogenic reversion of the KBR-1-p120 strain, the two calves in group G1 (inoculated with the KBR-1-p120 strain) were autopsied at 7 DPI, and the small intestines were emulsified. The emulsion (5 mL per dose) was then inoculated into two 7-day-old calves (group G2). These two calves were then autopsied 7 days later, and their small intestines were emulsified; 5 mL of this emulsion was then administered to two 7-day-old calves (group G3), and these calves were also autopsied at 7 DPI.

### 2.5. Detection of BCoV RNA via RT-PCR

Fecal and nasal samples from all calves used in the above experiment, as well as samples of their colon, duodenum, jejunum, and ileum, were subjected to RT-PCR as described previously [21]. RT-PCR targeting the N gene of BCoV was conducted using forward primer CTA GTA ACC AGG CTG ATG TCA ATA CC, reverse primer GGC GGA AAC CTA GTC GGA ATA, and probe FAM-CGC CTG ACA TTC TCG ATC-MGB [20]. RT-PCR was performed using the CFX Opus 96 model (Bio-Rad Laboratories, Inc., Hercules, CA, USA) and the following cycling conditions: reverse transcription at 45 °C for 30 min; activation of DNA polymerase at 95 °C for 10 min; 40 cycles of denaturation at 94 °C for 15 s; and annealing/elongation at 60 °C for 60 s.

## 3. Results and Discussion

### 3.1. Changes in Cell Morphology Induced by the KBR-1 Strain

The BCoV KBR-1 strain was isolated successfully after inoculating susceptible HRT-18 cells, and virus growth was confirmed in an IFA [17]; however, no CPE was observed under a bright field (BF) microscope, although some trypsin-induced damage was observed (Figure 1). A previous study reported that the KBR-1 strain grew to 10^3.0^ TCID_50_/mL at passage five, 10^4.2^ TCID_50_/mL at passage 20, and 10^6.2^ TCID_50_/mL at passage 40, in HRT-18 cells [17]; however, the maximum proliferation of the KBR-1 strain observed in the present study was <10^6.1^ TCID_50_/mL (10^6.1^ TCID_50_/mL at passage 50, 10^5.8^ TCID_50_/mL at passage 60, 10^5.6^ TCID_50_/mL at passage 70, and 10^5.5^ TCID_50_/mL at passage 79). To increase the virus titer, we inoculated the KBR-1 strain continuously onto susceptible MDBK cells from passage 80 to passage 120; the virus titer then increased to 10^7.7^ TCID_50_/mL (10^7.3^ TCID_50_/mL at passage 90, 10^7.6^ TCID_50_/mL at passage 100, 10^7.7^ TCID_50_/mL at passage 110, and 10^7.5^ TCID_50_/mL at passage 120). In addition, the KBR-1 strain exhibited CPEs in susceptible MDBK cells (Figure 1), which is consistent with the IFA results (Figure 1) obtained using a BCoV-specific fluorescent antibody. A recent BCoV isolation study inoculated MDBK cells with calf diarrhea samples obtained from the central part of Oromia, Ethiopia, and also demonstrated CPEs [22]. The MDBK cells began clumping after 24 h, and appeared thin and round after 48 h; the majority of the monolayer detached after 72 h [22]. Thus, MDBK cells show CPEs and allow the virus to propagate to high titers. Therefore, they are considered highly sensitive to BCoV infection, making them a useful tool for virus isolation.

### 3.2. Amino Acid Mutations in the Spike Genes of Passaged KBR-1 Strains

During serial passage (up to 70) in the HRT-18 cells, the S gene of the KBR-1-p70 strain acquired up to seven amino acid mutations (Table 1). These seven mutated amino acids (Q179R, K268R, Y554H, N714D, Q1118H, P1238S, and S1256L) were located in the S1 and S2 regions (Table 1). The KBR-1 strain (KBR-1-p80 to KBR-1-p120) inoculated into the MDBK cells harbored nine amino acid mutations at passage 80 and 13 at passage 90. These 13 amino acid mutations (S25A, L85R, D115G, P174R, Q179R, I367T, Y554H, T640I, N714D, L903S, P1238S, S1256L, and K1340N) persisted until passage 120 (Table 1). In general, the entry of coronaviruses into cells relies on specific interactions between S-trimers on the virion surface and host cell receptors [23]. The S ectodomain contains the viral attachment and entry subunit S1, as well as the membrane fusion subunit S2 [23]. For most coronaviruses, the N-terminal domain of the S1 subunit (S1-NTD; residues 15–298) recognizes cell surface carbohydrates, while the C-terminal domain (S1-CTD; amino acids 326–540) binds to specific protein receptors on the host cell [24]. The receptor recognized by the S1-NTD of BCoV is 5-N-acetyl-9-O-acetylneuraminic acid (Neu5, 9Ac2) [25], but recent studies suggest that it may act only as an attachment receptor for BCoV [26]. Another study suggests that the S1-CTD of BCoV strains contains a putative receptor binding domain that recognizes specific protein receptors [26]. BCoV comprises the S1-NTD and the S1-CTD protein [27]. It is assumed that the five mutations (S25A, L85R, D115G, P174R, and Q179R) in S1-NTD and the single mutation (I367T) in S1-CTD of the KBR-1-p120 strain (passaged for 90 to 120 generations) trigger changes in the receptor through which the virus binds to the cell surface.

### 3.3. Phylogenetic Tree of BCoV Strains

The ML tree of 106 BCoV complete S nucleotide sequences showed that all 12 KBR-1 strains (KBR-1-p10, -p20, -p30, -p40, -p50, -p60, -p70, -p80, -p90, -p100, -p110, and -p120) belonged to the GIIa group. The 12 KBR-1 strains were sequentially related to strains from the previous passage (Figure 2). A previous study suggested that the BCoV vaccine strain (BC94) that has been used for a long time in South Korea belongs to the GI genotype, and is only distantly related to the recent prevalent strains (all belonging to the GIIa genotype) [17]. Phylogenetic analysis also showed that the 12 KBR-1 strains (from KBR-1-p10 to KBR-1-p120) and the BC94 strain were the most distant from each other (Figure 2).

### 3.4. Clinical Signs/Symptoms and Virus Shedding by Infected Calves

The clinical scores for the two calves inoculated with the KBR-1-p10 strain (G4) were one (weak) and two (moderate) between 4 and 7 days post-inoculation. In addition, their appetite and activity decreased over the same time period (days 4–7) (Table 2). By contrast, calves in G1 (KBR-1-p120), G2 (inoculated with calf small intestine emulsion from G1), and G3 (inoculated with calf small intestine emulsion from G2) showed no symptoms of diarrhea or runny nose (Table 2). Their appetite and activity were normal (Table 2). The RT-PCR to detect BCoV RNA in nasal samples obtained at 7 days post-inoculation revealed that it was absent from some calf samples (Table 2). A previous study showed that WD-type BCoV can infect both the digestive and respiratory tracts of calves [15]; however, the KBR-1 strain used in the present study (an NCD-type BCoV) is believed to infect only the digestive tract.

BCoV RNA from feces samples collected from G4 showed a ct value of 24.8–33.9 for calf G4-1 and 25.1–34.2 for calf G4-2 (Table 2). Calf (G1-1) in the G1 group had BCoV RNA in its feces, with a ct value of 37.6–38.1, whereas calf G1-2 had a ct value of 36.2–37.5. However, a previous paper that performed RT-PCR using the primers and probes used in the present study defined a positive RT-PCR ct value < 35 [28]. Therefore, in the pathogenicity reversal experiment using the KBR-1-p120 strain, the ct values detected in samples from calves in the G1 group were almost negative. Additionally, the ct values in calves from the G2 and G3 groups were negative (Table 2). These data infer that the KBR-1-p120 strain was attenuated.

### 3.5. Organs from Calves Inoculated with BCoV

The two calves in G4 (inoculated with the KBR-1-p10 strain) were autopsied 7 days post-inoculation to detect BCoV RNA in the duodenum, ileum, and jejunum (Table 3). The ct value in the jejunum and ileum of G1-1 (inoculated with the KBR-1-p120 strain) was >35; the ct value in the ileum of calf G1-2 was also high (36.9) (Table 3). However, it was difficult to detect BCoV (KBR-1-p120) in the organs of the G1 group calves because the ct values were so high. A previous paper showed that when the mock group was challenged with virulent BCoV, the virus was retained in the organs, especially the duodenum, jejunum, and ileum [17]. In the mock group, the ct value in the duodenum was 31.8, that in the jejunum was 23.6–24.7, and that in the ileum was 26.2–25.5 [17]. Although the present study was performed using a small number of calves, after comparing the presence or absence of BCoV in the small intestine of autopsied calves, we can assume that the KBR-1-p120 strain was attenuated by the 120th passage. In the future, we plan to conduct an overdose safety clinical trial in pregnant cows to confirm the attenuation of the vaccine strain (KBR-1-p120 strain) for use as a live attenuated vaccine.

## 4. Conclusions

The BCoV KBR-1 strain was passaged continuously 120 times in HRT-18 cells and MDBK cells, resulting in 13 amino acid mutations in the spike gene. In MDCK cells, the titer of the BCoV (KBR-1 strain) increased, and the non-CPE type changed to a CPE type. When the KBR-1 strain (passage 120) was inoculated orally into calves that did not consume colostrum, almost no BCoV RNA was detected in their fecal samples or in the small intestine after autopsy. In addition, the pathogenic reversion test revealed that no calves showed clinical symptoms. Therefore, future experiments are planned to test the protective efficacy of this attenuated vaccine candidate strain (KBR-1-p120). First, we plan to inoculate pregnant cows with KBR-1-p120, and then to inoculate the newborn calves with virulent BCoV.

## Figures and Tables

**Figure 1 animals-14-00389-f001:**
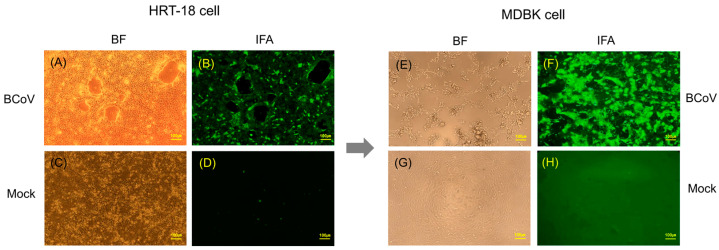
Non-CPE and CPE observed after inoculation of the BCoV KBR-1 strain onto HRT-18 and MDBK cells. BF (**A**) and IFA (**B**) images obtained at 4 days post-KBR-1 inoculation, and BF (**C**) and IFA (**D**) images obtained at 4 days post-mock inoculation, into HRT-18 cells. BF (**E**) and IFA (**F**) images obtained at 4 days post-inoculation of the KBR-1 strain, and BF (**G**) and IFA (**H**) images obtained at 4 days post-mock inoculation, into MDBK cells. BF = bright field; IFA = indirect fluorescence assay.

**Figure 2 animals-14-00389-f002:**
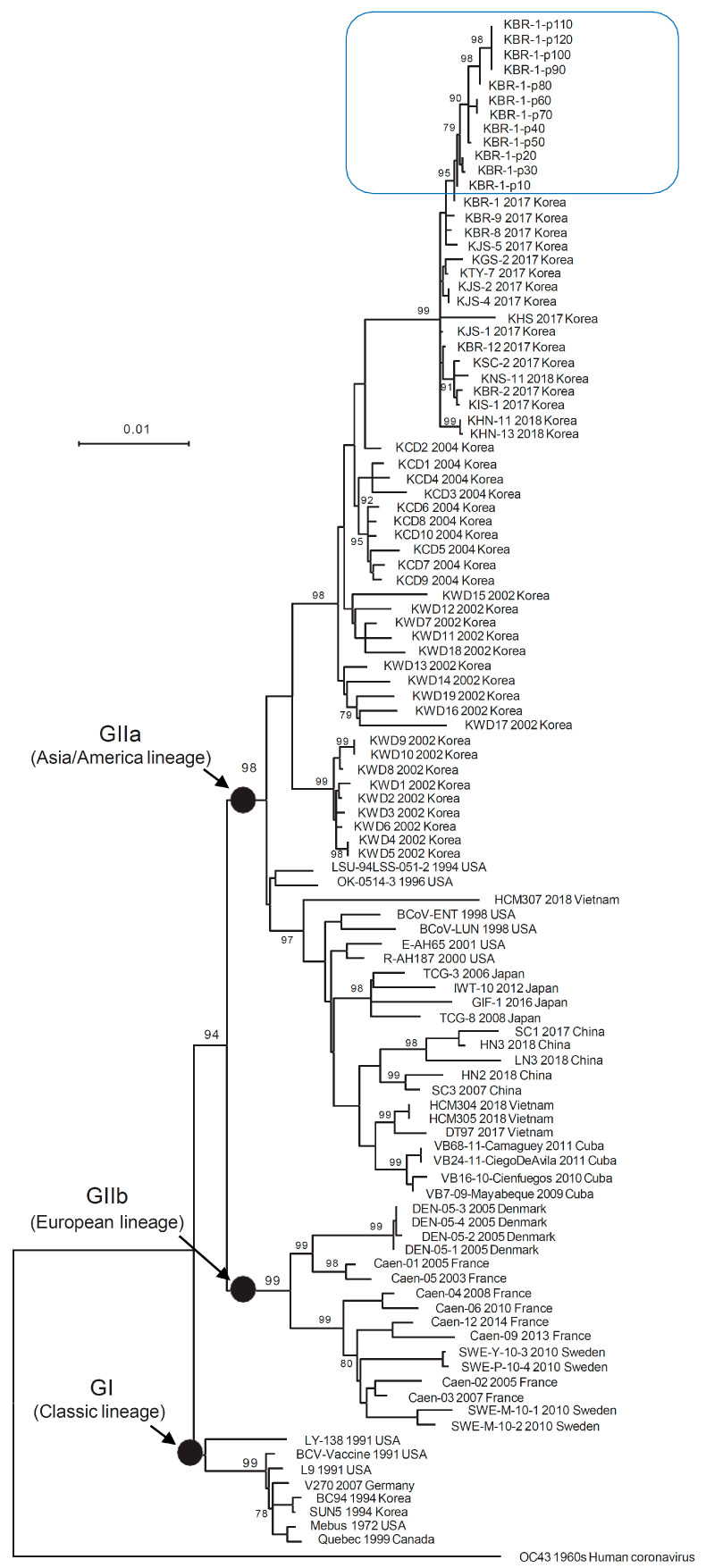
Maximum-likelihood phylogenetic tree (LogL = −14799.04) based on the complete S gene nucleotide sequences of 106 BCoV reference strains detected in Asian, North American, and European countries, including the 12 KBR-1 strains passaged in cells in this study. The phylogenetic tree was constructed using MEGA X software (the Tamura–Nei model and the nearest-neighbor interchange method). The blue box marks the 12KBR-1 strains.

**Table 1 animals-14-00389-t001:** Amino acid sequences of the spike gene of bovine coronavirus KBR-1 strains.

Host Cell	No. P *	Positions of Amino Acid Mutations in the Spike Gene of Bovine Coronavirus KBR-1 Strain															
		25	85	115	174	179	268	367	554	640	714	903	965	1118	1238	1256	1340
HRT-18	0	S	L	D	P	Q	K	I	Y	T	N	L	E	Q	P	S	K
	10								H								
	20					R			H				V				
	30					R			H				V		S		
	40					R			H		D				S	L	
	50					R			H		D				S	L	
	60					R	R		H		D			H	S	L	
	70					R	R		H		D			H	S	L	
MDBK	80	A	R	G		R		T	H		D				S	L	
	90	A	R	G	R	R		T	H	I	D	S			S	L	N
	100	A	R	G	R	R		T	H	I	D	S			S	L	N
	110	A	R	G	R	R		T	H	I	D	S			S	L	N
	120	A	R	G	R	R		T	H	I	D	S			S	L	N

* No. P: number of passages in cell culture.

**Table 2 animals-14-00389-t002:** Clinical scores and virus shedding (up to 7 days post-BCoV inoculation).

Group	No. Calve	Symptom	Clinical Score/RNA Copy Number (ct Value) of BCoV by DPI *							
			0	1	2	3	4	5	6	7
G1 (KBR-1-p120, 10^7.0^ TCID_50_/mL/dose)	G1-1	Diarrhea	0/-	0/-	0/-	0/-	0/37.6	0/-	0/38.1	0/-
		Nasal	0/-	0/-	0/-	0/-	0/-	0/-	0/-	0/-
		Appetite	0	0	0	0	0	0	0	0
		Activity	0	0	0	0	0	0	0	0
	G1-2	Diarrhea	0/-	0/-	0/-	0/-	0/37.5	0/36.2	0/-	0/-
		Nasal	0/-	0/-	0/-	0/-	0/-	0/-	0/-	0/-
		Appetite	0	0	0	0	0	0	0	0
		Activity	0	0	0	0	0	0	0	0
G2 (Small intestine emulsion from G1, 5 mL/dose)	G2-1	Diarrhea	0/-	0/-	0/-	0/-	0/-	0/-	0/-	0/-
		Nasal	0/-	0/-	0/-	0/-	0/-	0/-	0/-	0/-
		Appetite	0	0	0	0	0	0	0	0
		Activity	0	0	0	0	0	0	0	0
	G2-2	Diarrhea	0/-	0/-	0/-	0/-	0/-	0/-	0/-	0/-
		Nasal	0/-	0/-	0/-	0/-	0/-	0/-	0/-	0/-
		Appetite	0	0	0	0	0	0	0	0
		Activity	0	0	0	0	0	0	0	0
G3 (Small intestine emulsion from G1; 5 mL/dose)	G3-1	Diarrhea	0/-	0/-	0/-	0/-	0/-	0/-	0/-	0/-
		Nasal	0/-	0/-	0/-	0/-	0/-	0/-	0/-	0/-
		Appetite	0	0	0	0	0	0	0	0
		Activity	0	0	0	0	0	0	0	0
	G3-2	Diarrhea	0/-	0/-	0/-	0/-	0/-	0/-	0/-	0/-
		Nasal	0/-	0/-	0/-	0/-	0/-	0/-	0/-	0/-
		Appetite	0	0	0	0	0	0	0	0
		Activity	0	0	0	0	0	0	0	0
G4 (KBR-1-p10, 10^5.0^ TCID_50_/mL/dose)	G4-1	Diarrhea	0/-	0/-	0/-	0/29.5	1/27.4	1/24.8	2/28.6	1/33.9
		Nasal	0/-	0/-	0/-	0/-	0/0	0/0	0/0	0/0
		Appetite	0	0	0	0	1	0	1	0
		Activity	0	0	0	0	0	0	1	0
	G4-2	Diarrhea	0/-	0/-	0/-	0/-	1/30.8	2/27.2	2/25.1	1/34.2
		Nasal	0/-	0/-	0/-	0/-	0/0	0/-	0/-	0/-
		Appetite	0	0	0	0	1	1	1	0
		Activity	0	0	0	0	0	1	1	0

* DPI: days post-inoculation.

**Table 3 animals-14-00389-t003:** Detection of virus in enteric organs of calves inoculated with BCoV KBR-1.

Organ	RNA Copy Number (ct Value) of BCoV by DPI *							
	G1-1	G1-2	G2-1	G2-2	G3-1	G3-2	G4-1	G4-2
Large intestine	- **	-	-	-	-	-	-	-
Duodenum	-	-	-	-	-	-	27.9	25.3
Jejunum	37.4		-	-	-	-	20.5	18.8
Ileum	38.2	36.9	-	-	-	-	19.6	20.1

* DPI: days post-inoculation, **: not detected.

## Data Availability

The complete spike gene sequences of the 12 BCoV strains (accession numbers: OR909655-OR909666) passaged using HRT-18 and MDBK cells have been deposited in GenBank.

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
