# Peer review of "Characterization and Spike Gene Analysis of a Candidate Attenuated Live Bovine Coronavirus Vaccine"

_animals, 2024, doi:10.3390/ani14030389_

Round 1

Reviewer 1 Report

Comments and Suggestions for Authors

The authors aim to characterize the BCoV KBR-1 strain they attenuated by passing it in HRT-18 and MDBK cells. They perform sequencing of the S-protein of the virus and investigate the pathogenicity in infection experiments in immunologically naive calves.  

General concept comments

These comments are focused on the scientific content of the manuscript and should be specific enough for the authors to be able to respond.

ll 59-60: Is it feasible to (retrospectively?) find suitable numbers of agricultural producers who either apply or do not apply GI-based vaccines and compare the levels of calf diarrhea at the respective sites, thus checking for statistically significant correlations in regard to GI-based protection from GIIa type corona viruses?

L109: Which methods were used to demonstrate the calves were free from BCoV antigens and antibodies?

Ll111-126: What was the genetic background of the calves used?
Did all calves belong to the same breed? 
Have differences/polymorphisms of bovine S-protein receptors associated with higher or lower susceptibility been published?
Why did you choose to perform the experiments using only 2 biological replicates/group? How did you calculate this number? What is the variance in BCoV-infections, specifically in developing symptoms in immunologically naïve calves?
How did you ensure calves did not encounter other pathogens during the study?

Ll252-254: This data would indeed be most interesting investigating whether KBR-1-p120 would be a suitable vaccine candidate.
Why did you decide against including this experiment into the present study?

Missing Methods: consider providing cell culture media and additives, in which virus was grown in the respective cells.

Missing Methods: how did you concentrate the virus for the inoculation of the calves?

G2 and G3 experiments: is it plausible to assume the virus did not regain pathogenicity, if you infect calves with emulsified intestines showing a very low ct value?
Is it feasible to show virus assembly indeed occurs in the G1-calves, for example using immunohistochemistry?

Specific comments
l 55: Sentence unclear, probably should read as: BCoV is involved in digestive and respiratory system disorders?

Consider renaming the calves to only include the G-group and the number for a better readability. Thus you could refer to P3-301 and P3-302 as G3-1 and G3-2, respectively.

Please consider weather both fig.2 and fig.3 are necessary

Comments on the Quality of English Language

The quality of English language is very good.

Author Response

Reviewer 1

The authors aim to characterize the BCoV KBR-1 strain they attenuated by passing it in HRT-18 and MDBK cells. They perform sequencing of the S-protein of the virus and investigate the pathogenicity in infection experiments in immunologically naive calves.  

General concept comments

These comments are focused on the scientific content of the manuscript and should be specific enough for the authors to be able to respond.

Comment 1. Lines 59-60: Is it feasible to (retrospectively?) find suitable numbers of agricultural producers who either apply or do not apply GI-based vaccines and compare the levels of calf diarrhea at the respective sites, thus checking for statistically significant correlations in regard to GI-based protection from GIIa type corona viruses?

Answer: We are conducting a 1-year survey (from December 2023 to December 2024) to statistically analyze the occurrence of the GIIa type (BCoV) on cow farms that use the GI type (BCoV) vaccine. Therefore, we expect that future papers will show a clearer association between use of the GI type (BCoV) vaccine and prevalent GIIa type (BCoV) strains.

Comment 2. L109: Which methods were used to demonstrate the calves were free from BCoV antigens and antibodies?

Answer: We have clarified this by revising the text as follows:

“7-day-old calves (negative for BCoV antigen by RT-PCR [20] and antibodies by serum neutralization antibody test [17]) that did not consume colostrum were assigned to two groups (G1 and G4).” (lines: 109–111).

Comment 3. Ll111-126: What was the genetic background of the calves used?
Did all calves belong to the same breed? 
Have differences/polymorphisms of bovine S-protein receptors associated with higher or lower susceptibility been published?
Why did you choose to perform the experiments using only 2 biological replicates/group? How did you calculate this number? What is the variance in BCoV-infections, specifically in developing symptoms in immunologically naïve calves?
How did you ensure calves did not encounter other pathogens during the study?

Answer: There are two representative breeds of cattle (Korean beef and dairy cattle) in Korea, and they show the same susceptibility to bovine virus diseases. All calves used in the experiments were dairy cattle (Holstein).

To prevent cross-contamination, experiments on each group were performed in different compartments. Additionally, the experimenters adhered to quarantine guidelines and conducted experiments without crossover.

In addition, clinical symptoms were with the clinical scores of the control group to confirm whether pathogenicity was weakened. The calf head count used for the pathogenicity reversion test was conducted according to WOAH guidelines (Manual of Diagnostic tests and vaccines for Terrestrial animals, 12th edition 2023, Chapter 1.1.8 Principle of veterinary vaccine production).

Comment 4. Ll252-254: This data would indeed be most interesting investigating whether KBR-1-p120 would be a suitable vaccine candidate.
Why did you decide against including this experiment into the present study?

Answer: The main purpose of the paper was to confirm the potential of BCV KBR-1 (P120) as a live vaccine candidate by confirming whether the pathogenicity of KBR-1 (P120) is attenuated, and whether pathogenicity is recovered after reverse passage. Therefore, in future papers, we plan use the BCV KBR-1(P120) strain to conduct immunogenicity experiments in 4-month-old calves and pregnant cows.

Comment 5. Missing Methods: consider providing cell culture media and additives, in which virus was grown in the respective cells.

Answer: Thank you. We have revised the text as follows:

“The KBR-1 strain was propagated by inoculation to the multiplicity of infection (MOI) 0.1 onto monolayers of human rectal tumor (HRT-18) cells in RPMI 1640 (Gibco, Grand Island, NY, USA) containing 5 μg/ml trypsin (Gibco, Grand Island, NY, USA) and then cultured 37°C/5% CO2 for 4 days. Additionally, to increase the titer of the KBR-1 strain, it was inoculated to the multiplicity of infection (MOI) 0.1 onto Madin–Darby bovine kidney (MDBK) cells in α-MEM (Gibco, Grand Island, NY, USA) containing 5 μg/ml trypsin (Gibco, Grand Island, NY, USA) and cultured at 37°C/5% CO2 for 4 days to observe cytopathic effects (CPE).” (lines: 71–79).

Comment 6. Missing Methods: how did you concentrate the virus for the inoculation of the calves?

Answer: We have revised the text as follows:

“KBR-1-p10 (concentrated by the density gradient ultracentrifugation due to the low titer (103.5-4.0 TCID50/mL))” (lines 113–114).

Comment 7. G2 and G3 experiments: is it plausible to assume the virus did not regain pathogenicity, if you infect calves with emulsified intestines showing a very low ct value?

Answer: Pathogenicity reversion experiments are generally conducted sequentially for up to five passages using emulsions from animals that were first inoculated according to WOAH guidelines. To regain pathogenicity, virus that is not completely attenuated shows both a high titer and pathogenicity when infecting animals repeatedly. Therefore, our experiments show complete attenuation, with minimal viral shedding and virtually no viral presence in the organs.

Comment 8. Is it feasible to show virus assembly indeed occurs in the G1-calves, for example using immunohistochemistry?

Answer: Unfortunately, real-time PCR signals in the organs of calves in the G1 group were weak, and virus was not confirmed by immunohistological staining; therefore, we did not include these data.

Specific comments
Comment 9. l 55: Sentence unclear, probably should read as: BCoV is involved in digestive and respiratory system disorders?

Answer: We have revised the text as follows:

“Additionally, through animal experiments, it has been confirmed that BCV, the cause of WD, can induce pathogenic effects in both the digestive and respiratory systems of calves.” (lines: 54–56).

Comment 10. Consider renaming the calves to only include the G-group and the number for a better readability. Thus you could refer to P3-301 and P3-302 as G3-1 and G3-2, respectively.

Answer: We have revised all of the animal numbers in the manuscript text, and in Tables 2 and 3.

Comment 11. Please consider weather both fig.2 and fig.3 are necessary

Answer: We have removed Figure 3.

Reviewer 2 Report

Comments and Suggestions for Authors

Bovine coronavirus (BCoV) is a major pathogen that causes neonatal calf diarrhea (NCD), winter dysentery (WD), and respiratory disease in calves. Since the early 2000s, WD and NCD caused by BCoV have been prevalent in Korea, indicating a shift to the GIIa genotype with potential antigenic differences from the current vaccine. With BCoV genotypes evolving, there is an urgent need for vaccines against the currently prevalent strains.

The authors of this study aimed to develop a live attenuated vaccine from a South Korean BCoV isolate. They passed the isolated KBR-1 for 79 passages on HRT-18 cells and continued to pass it on MDBK cell to 120 passages. The KBR-1-p120 strain demonstrates high viral titer and cytopathogenic effects in MDBK cells. The KBR-1-p120 strain contains 13 amino acid mutations in the spike gene. This genetic alteration is a crucial aspect of the strain's characterization and attenuation. Then, the authors orally inoculated Calves with a less passaged isolate (KBR-1-p10) and an attenuated candidate strain (KBR-1-p120) and demonstrated that KBR-1-p120 was attenuated in calf.

The study was carefully designed and carried out. The resulting KBR-1-p120 was attenuated and likely to be a potential attenuated BCoV vaccine candidate. However, a limitation of the study lies in the small number of animals included in animal study. Additionally, further evaluation is needed to assess the immunogenicity of the attenuated KBR-1-p120. Furthermore, there are some concerns regarding the manuscript need to be addressed. 

1.      It would be beneficial to include more details about the passaging process in the Materials and Methods section, such as the Multiplicity of Infection (MOI) or the volume used for passaging.

2.      On Line 118, clarification is needed regarding the nature of the samples. Are they swabs or actual feces and nasal fluid? Additionally, a description of the sample preparation method for real-time RT-PCR would enhance clarity.

3.      In Figure 1, the scale bars are not clearly visible, and their lengths are not specified in the legend. Moreover, Figure 1F lacks sufficient clarity.

4.      Between Line 118 and 125, it is crucial to provide information about the virus titer in the small intestine emulsion used for the pathogenic reversion test. Without knowledge of the virus titer, it becomes challenging to assess the potential pathogenic reversion of the virus.

Author Response

Reviewer 2

Bovine coronavirus (BCoV) is a major pathogen that causes neonatal calf diarrhea (NCD), winter dysentery (WD), and respiratory disease in calves. Since the early 2000s, WD and NCD caused by BCoV have been prevalent in Korea, indicating a shift to the GIIa genotype with potential antigenic differences from the current vaccine. With BCoV genotypes evolving, there is an urgent need for vaccines against the currently prevalent strains.

The authors of this study aimed to develop a live attenuated vaccine from a South Korean BCoV isolate. They passed the isolated KBR-1 for 79 passages on HRT-18 cells and continued to pass it on MDBK cell to 120 passages. The KBR-1-p120 strain demonstrates high viral titer and cytopathogenic effects in MDBK cells. The KBR-1-p120 strain contains 13 amino acid mutations in the spike gene. This genetic alteration is a crucial aspect of the strain's characterization and attenuation. Then, the authors orally inoculated Calves with a less passaged isolate (KBR-1-p10) and an attenuated candidate strain (KBR-1-p120) and demonstrated that KBR-1-p120 was attenuated in calf.

The study was carefully designed and carried out. The resulting KBR-1-p120 was attenuated and likely to be a potential attenuated BCoV vaccine candidate. However, a limitation of the study lies in the small number of animals included in animal study. Additionally, further evaluation is needed to assess the immunogenicity of the attenuated KBR-1-p120. Furthermore, there are some concerns regarding the manuscript need to be addressed. 

Comment 1.      It would be beneficial to include more details about the passaging process in the Materials and Methods section, such as the Multiplicity of Infection (MOI) or the volume used for passaging.

Answer: Thank you. We have revised the text as follows:

“The KBR-1 strain was propagated by inoculation to the multiplicity of infection (MOI) 0.1 onto monolayers of human rectal tumor (HRT-18) cells in RPMI 1640 (Gibco, Grand Island, NY, USA) containing 5 μg/ml trypsin (Gibco, Grand Island, NY, USA) and then cultured 37°C/5% CO2 for 4 days. Additionally, to increase the titer of the KBR-1 strain, it was inoculated to the multiplicity of infection (MOI) 0.1 onto Madin–Darby bovine kidney (MDBK) cells in α-MEM (Gibco, Grand Island, NY, USA) containing 5 μg/ml trypsin (Gibco, Grand Island, NY, USA) and cultured at 37°C/5% CO2 for 4 days to observe cytopathic effects (CPE).” (lines: 71–79).

Comment 2.     On Line 118, clarification is needed regarding the nature of the samples. Are they swabs or actual feces and nasal fluid? Additionally, a description of the sample preparation method for real-time RT-PCR would enhance clarity.

Answer: We have revised the text as follows:

“Additionally, fecal and nasal samples were collected daily for 7 days post-inoculation (DPI). There samples was collected using sterilized cotton swabs, placed in liquid transport medium, vortexed, and stored in a deep freezer at -70°C until real-time PCR.” (lines: 120–124).

Also, the real-time PCR method is explained in section "2.5. Detection of BCoV RNA by RT-PCR."

Comment 3.     In Figure 1, the scale bars are not clearly visible, and their lengths are not specified in the legend. Moreover, Figure 1F lacks sufficient clarity.

Answer: We have revised Figure 1 as suggested.

Comment 4.      Between Line 118 and 125, it is crucial to provide information about the virus titer in the small intestine emulsion used for the pathogenic reversion test. Without knowledge of the virus titer, it becomes challenging to assess the potential pathogenic reversion of the virus.

Answer: We have revised the text accordingly.

In the pathogenic reversal test, the small intestine emulsion from calves in groups G1, G2, and G3 was cultured with cells, but no virus was detected. However, the average Ct value of the RT-PCR was about 37.6 only in the small intestine emulsion from G1 calves; nothing was amplified from the small intestine emulsion from G2 and G3 calves. These results confirm that since the KBR-1-p120 strain was attenuated, the KBR-1-p120 strain is safe during animal passage, even when the pathogenicity reversion test is performed using a high titer (107.0 TCID50/mL/dose).

Round 2

Reviewer 1 Report

Comments and Suggestions for Authors

Comment 7: According to the TAS Biologicals annexes, "When the organism is not recovered from any intermediate in vivo passage, a reasonable attempt should be made to repeat the test in 10 animals (90% probability of isolating the organism at 20% probability of recovery – see Appendix) using in vivo passaged material from the last passage in which the organism was recovered." And "If the target organism is not recovered, the experiment is considered to be completed with the conclusion that the target organism does not show an increase in or reversion to virulence."

Do you have stocks of G1/G2/G3 emulsions preserved, which could be used to try propagating the virus in vitro? 

  • Target Animal Safety - Examination of Live Veterinary Vaccines in Target Animals for Absence of Reversion to Virulence - Annexes
    VICH GL41 (TAS Biologicals) July 2007 - Implemented in July 2008
  • Target animal safety (vichsec.org)

Author Response

Reviewer 1

Comment 7: According to the TAS Biologicals annexes, "When the organism is not recovered from any intermediate in vivo passage, a reasonable attempt should be made to repeat the test in 10 animals (90% probability of isolating the organism at 20% probability of recovery – see Appendix) using in vivo passaged material from the last passage in which the organism was recovered." And "If the target organism is not recovered, the experiment is considered to be completed with the conclusion that the target organism does not show an increase in or reversion to virulence."

Do you have stocks of G1/G2/G3 emulsions preserved, which could be used to try propagating the virus in vitro? 

Answer: To measure the virus titer of G1/G2/G3 emulsions, MDBK cells were inoculated. And the cells inoculated with G1/G2/G3 emulsions were measured by CPE, IFA, and real-time PCR, but BCoV was all confirmed negative. It was confirmed that BCoV does not exist in G1/G2/G3 emulsions as the virus did not proliferate even after repeated experiments in MDBK cells.

We were also asked a similar question by Reviewer 2 and responded as follows: “In the pathogenic reversal test, the small intestine emulsion from calves in groups G1, G2, and G3 was cultured with cells, but no virus was detected. However, the average Ct value of the RT-PCR was about 37.6 only in the small intestine emulsion from G1 calves; nothing was amplified from the small intestine emulsion from G2 and G3 calves. These results confirm that since the KBR-1-p120 strain was attenuated, the KBR-1-p120 strain is safe during animal passage, even when the pathogenicity reversion test is performed using a high titer (107.0 TCID50/mL/dose).”